# COVID-19 Stressors and Aggression among Chinese College Students: The Mediation Role of Coping Strategies

**DOI:** 10.3390/ijerph20043171

**Published:** 2023-02-10

**Authors:** Wencai Hu, Mengru Sun

**Affiliations:** 1College of Humanities and Communication, Zhejiang University of Finance and Economics, Hangzhou 310018, China; 2College of Media and International Culture, Zhejiang University, Hangzhou 310058, China

**Keywords:** COVID-19, stressors, coping strategies, aggression, college students

## Abstract

Although college students experienced excessive stressors (COVID-19 disease and negative COVID-19 news) during the COVID-19 pandemic, few studies have been aimed at coping strategies used by college students to deal with stress caused by the pandemic. Coping strategies are efforts to deal with anxiety in the face of a perceived threat or stress. Aggression is harmful social interaction with the intention of inflicting damage or harm upon another individual. In the present study, we aimed to examine the direct effect of stressors resulting from the pandemic on college students’ aggression, as well as the indirect effect via their coping strategies. Through a cross-sectional survey of 601 Chinese college students (M-age = 20.28), we tested the proposed framework. We first found that information stressors of COVID-19 ranked highest among the four stressors of the pandemic. Results also indicated that college students’ stressors of COVID-19 were directly and positively associated with their aggressive behavior. For the indirect effect, college students would adopt both adaptive coping strategies (self-help strategy) and maladaptive coping strategies (avoidance strategy and self-punishment strategy) with the stressors of COVID-19. Furthermore, adaptive coping strategy (approach strategy) was negatively related to their aggression, whereas maladaptive coping strategy (avoidance strategy and self-punishment strategy) was positively related to their aggressive behavior. The present research extends the general strain theory in the COVID-19 context. Practical implications are also discussed.

## 1. Introduction

A set of factors have been verified to be associated with adolescents’ aggression and campus violence, whereby an important predictor is stress [1,2]. Stress could be induced by events in daily life and emergencies, such as the COVID-19 pandemic [3]. During the pandemic, people all around the world have been under pressure from work, study, life, and the economy [4,5]. The COVID-19 pandemic had a far-reaching impact on the education community for both teachers and learners [6]. College students in China, in particular, have been forced to take online remote courses for several semesters. During this period, many students cannot return to school, get along with classmates and friends, or conduct their studies normally [7]. Although most courses have now resumed face-to-face teaching, a series of problems such as delays in academic progress and employment difficulties caused by the pandemic are still testing college students. In China’s educational environment, college students who have just separated from family supervision often experience social maladjustment [8,9]. Studies show that college students in China are a group with frequent psychological distress and social problems [10,11].

In addition to the immediate health concerns associated with COVID-19, a range of stressors related to the COVID-19 pandemic have therewith raised prominent concerns about the mental health and well-being of college students [12]. For example, in the face of tremendous pressure and emergencies related to COVID-19, college students’ aggressive behavior may also directly increase (e.g., school bullying). Therefore, it is important to theorize the adverse effects of stressors associated with COVID-19 on college students’ aggressive behavior. Aggression, an important social issue, occurs frequently among adolescents around the world [13]. Because some aggressive incidents have a serious social impact, they draw concerns from all walks of life. Aggressive behavior deliberately causes harm or threats to the body, mind, or property. For adolescents and young adults who are experiencing rapid physical and mental growth, aggression brings them not only direct physical harm, but also psychological trauma [14]. Therefore, it is urgent to conduct research on the causes of aggression and preventive measures to strengthen prevention and minimize occurrences of aggressive behavior among adolescents.

Although a large number of studies have confirmed the moderating effect of coping between the association of stressors and stress consequences, relatively few researchers have explored the mediating effects of different coping styles and strategies in this relationship [15,16]. Therefore, in this study we intend to utilize general strain theory and frustration-aggression theory to understand the mechanism of stressors related to COVID-19 on Chinese college students in the context of COVID-19. During the COVID-19 pandemic, college students have fewer opportunities to access and seek help from both schools and social networks but face higher risk of online abuse than before [17]. Hence, it becomes more possible for college students to take maladaptive coping strategies along with problem behaviors such as aggressive behavior with the various COVID-19 related stressors. Hence, the first aim of the present study is to explore the levels of different stressors during the pandemic. The second aim is to investigate if the stressors of COVID-19 experienced by college students are significantly associated with aggression and to examine the potential mediating role of coping strategies in this association.

### 1.1. Stressors of COVID-19 and Adolescents’ Aggression

Aggression refers to violent attacks or threats by an individual intentionally aimed at other individuals [18]. These violent behaviors cause significant problems across the world [19,20], and China is no exception with its own high rates of violent behaviors [21]. Violent incidents can cause serious negative outcomes to the victims, such as deteriorating school performance, distress, depression, and suicide [22].

Stress refers to a state of physical and mental tension caused by a series of internal and external stimuli [2]. Appropriate stress is beneficial to maintain the individual’s physical and mental health and help individuals adapt to the surrounding environment, but excessive pressure can reduce college students’ ability to adapt to school. For example, stress could cause adverse effects on the body and mind, leading to anxiety, sleep disorders, and other psychological problems [21]. Stress also has a negative impact on the establishment and maintenance of interpersonal relationships, triggering or exacerbating loneliness [20]. While stressors would induce a set of stress consequences, scholars divide stress into three dimensions: behavioral, psychological, and physical symptoms [23]. For behavioral consequences, previous theories have shown that stress plays an important role in the development of an individual’s aggression. For example, the frustration–aggression theory states that people are prone to aggressive behavior when frustrated [24].

Many empirical researchers also verified the relationship between stress and aggression among adolescents and young adults [25,26], wherein college students who experienced more stress showed a high level of aggression [27]. Furthermore, stress and aggression are of particular concern among certain groups (i.e., urban youth and female university students) and courses (i.e., physical education) [28,29,30]. Specifically, among Chinese college students, stress levels of more aggressive individuals were significantly higher than those of less aggressive individuals [31]. During the period of the COVID-19 pandemic, research has found that COVID-19 stress was significantly and positively related to physical and psychological intimate partner aggression perpetration along with higher intimate partner aggression rates [32,33].

It was also reported that college students had experienced extensive and heightened anxiety and concerns due to the lockdown measures [34]. In consequence, college students had to displace from university campuses and friends, experienced concern about health for themselves and their families, faced technology dependence and insufficient digital literacy, and experienced depression about academic and financial affairs [30,35,36,37,38]. According to the frustration-aggression model, the frustrating lockdown and social distancing experiences and stressors during COVID-19 would lead to college students’ aggression because a set of these previous basic goals were blocked [39].

### 1.2. The Mediation Role of Coping Strategies

Research suggest that the incapacity of adolescents to withstand frustration under pressure may induce the occurrence of violence [28,29,30]. College students, exposed to fierce social competition, are faced with pressures from family, society, and learning [30]. The United Nations Children’s Fund [40] has investigated the educational conditions of various countries and found that school life puts many kinds of unhealthy pressures on adolescents, including studies, employment, and economics. The sudden health incidents resulting from the COVID-19 pandemic were intertwined with previous pressures, making college students feel overwhelmed. During this period, if students could not deal with various setbacks appropriately, they were prone to commit acts of violence against others to vent the troubles caused by their stress (i.e., acts against others) [41].

The general strain theory effectively explains why some individuals under stress would exhibit socially problematic behaviors [42]. Agnew [42] states that the likelihood of individuals to adopt problem behaviors to cope with stress mainly depends on their ability and opportunity to use legal means to relieve stress, problem behavior costs, social connections, and problem behavior tendencies. Among these factors the ability to legally relieve stress includes personal cognition and problem-solving ability. From this perspective, coping reflects the ability of an individual to relieve stress. As suggest by Agnew [42], stress is most likely to result in delinquency when “the constraints to delinquent coping are low and the constraints to nondelinquent coping are high, and the adolescent has a disposition for delinquent coping.” Coping is one of the critical mediating factors in the relationship between psychological stress and related consequences [43]. A large amount of empirical evidence also shows that adolescents’ responses to stress, including their coping styles, will lead to different consequences [44,45].

Coping is a continuous and dynamic process, and coping style is a habitual strategy used by individuals to deal with and manage stressful life events and problems [45,46]. In fact, coping styles and strategies are essential in adolescence because the whole process requires adolescents to constantly face changes. This stage is the easiest time to start adapting a person’s mental health or social behavior to a particular pattern [47]. The coping strategies of adolescents affect their development and their adaptation to the social environment, which can be negative or positive [48,49,50]. In particular, adaptive coping styles and strategies can alleviate problem behaviors, while maladaptive coping styles and strategies can increase problem behaviors. For example, both adaptive and maladaptive cognitive coping are associated with the quality of life of homeless young adults [51]. Unfortunately, many adolescents often adopt negative or maladaptive coping strategies to deal with stress, leading to multifarious types of stress consequences [49].

Therefore, stress may indirectly affect college students’ aggressive behavior through their coping strategies. As suggested by general strain theory, along with coping strategies, many factors can regulate the relationship between stress and social problem behaviors, such as family economic status, involvement in orthodox activities, personal beliefs, community environment, and peers with whom they interact [42]. Specifically for coping strategies, researchers believe that individuals adopt coping strategies to solve personal and interpersonal problems and try to master, minimize, or tolerate stress and conflict [52]. These coping strategies also serve as prospective predictors of certain adjustments and well-being indices, including approach, self-help, accommodation, avoidance, and self-punishment [53]. However, previous research has indicated that adopting dysfunctional coping strategies to deal with stress can lead to various maladaptive behaviors, including depression, eating disorders, and substance abuse disorders [54,55,56]. For example, students who were diagnosed as being disturbed were all characterized by a high degree of withdrawal coping strategy, which would lead to adolescents’ negative health outcomes, such as anxiety, delinquency, and drug addiction [57].

Despite the fact that Chinese college students may have endured excessive stressors during the pandemic, few studies are aimed at college student groups [58] and their coping strategies under the stress of a health emergency such as a pandemic [59]. For example, some researchers found that under a series of stressors caused by COVID-19, some Chinese college students did not adopt avoidance coping strategies but tended to solve problems. However, other college students showed negative psychological reactions, such as panic, anxiety, guilt, and dissatisfaction [60,61,62,63].

### 1.3. Purpose of the Study

During the COVID-19 pandemic, studies indicated that the rates of bullying involvement were far higher than before the pandemic, including a variety of bullying forms (i.e., physical and verbal bullying) [64]. However, previous research mainly either analyzed the violence of general populations with diverse demographic characteristics or intimate partner violence. The present study fills the research gaps by focusing on college student groups, which share more demographic homogeneity and a unique experience of pandemic lockdown. This group has a similar age, lifestyle, and finances that may benefit further policy suggestions for the university community.

The main purpose of this study is thus to examine the association between stressors of the COVID-19 pandemic and Chinese college students’ aggression, with a particular focus on the five subdimensions of the coping strategies: self-help, approach, accommodation, avoidance, and self-punishment. Specifically, we intend to examine the direct effect of the stressor of the COVID-19 pandemic on college students’ aggression, as well as the indirect effect through their different coping strategies (see Figure 1). We expect the findings of the present study could shed light on fostering caring interpersonal relationships at universities for education and health practitioners.

**Research Question 1.** Which stressors are the most prominent stressors of the COVID-19 pandemic?

**Hypothesis** **1.***College students’ perceived stressors of COVID-19 are positively related to their aggressive behavior*.

**Hypothesis** **2.***College students’ perceived stressors of COVID-19 are positively related to (a) self-help strategy, (b) approach strategy, (c) accommodation strategy, (d) avoidance strategy, and (e) self-punishment strategy*.

**Hypothesis** **3.***College students’ adaptive coping strategies (i.e., (a) self-help strategy, (b) approach strategy, and (c) accommodation strategy) are negatively related to their aggressive behavior, whereas maladaptive coping strategies (i.e., (d) avoidance strategy and (e) self-punishment strategy) are positively related to their aggressive behavior*.

## 2. Materials and Methods

### 2.1. Participants

College students were recruited from a university in Zhejiang Province, mainland China. The survey was conducted in 2020, and 613 students participated. The students voluntarily participated in the study from courses taught by the researcher. We obtained consent from each student before the survey. The survey lasted approximately a month. Five participants were excluded owing to missing answers or because they did not take adequate time to answer the questions. The remaining final valid questionnaires were 601.

Participants’ ages ranged from 16 to 26 years (M = 20.28, SD = 1.72). Most of the participants were girls (*n* = 463, 77.0 %). The majority of them identified as Han ethnicity (*n* = 552, 91.8 %). Most of them reported that their family socioeconomic status was middle (*n* = 414, 68.9 %), whereas the number of people who stated their family socioeconomic status was upper-middle and lower-middle was similar (*n* = 69, 11.5 %; *n* = 101, 16.8 %, respectively). Only 14 students considered their family socioeconomic status was very low, and 3 students thought their family status was very high. As for academic performance, 48.1% of the students (*n* = 289) reported their academic performance was average; 34.9% (*n* = 210) reported their academic performance was good; 14.5% (*n* = 87) reported their academic performance was poor or very poor; and 2.5% (*n* = 15) reported their academic performance was very good.

### 2.2. Measures

#### 2.2.1. Stressors of COVID-19

The COVID-19 stressors was measured by adapting from Ye et al. [23]. Respondents were asked to answer on a 5-point Likert scale, “Talk about the following aspects of the pressure you feel during the COVID-19 pandemic?” The scales included four items: disease stressors of COVID-19 (“I am worried that I will be infected by COVID-19”), information stressors of COVID-19 (“I heard some negative news about COVID-19”), measures stressors of COVID-19 (“Academic schedule was disrupted”), and environmental stressors of COVID-19 (“I am separated and alienated from my classmates and friends”), anchored by 5 = “great stress” and 1 = “no stress”. In our study, Cronbach’s α = 0.725, and greater scores indicated a higher level of COVID-19-related stress (M = 2.57, SD = 0.79).

#### 2.2.2. Coping Strategies

Coping strategies were measured by adapting from Zuckerman and Gagne [53], comprising five subdimensions. Respondents were asked to answer on five 5-point Likert scales, “To what extent do you use the following strategies to relieve stress?” (anchored by 5 = “always use this coping strategy” and 1 = “never use this coping strategy”). The five subscales included self-help (“I try to get emotional support from family and friends”), accommodation (“I accept the reality of the fact it happened”), approach (“I do what has to be done, one step at a time”), avoidance (“I pretend that it has not really happened”), and self-punishment (“I punish myself”). In addition, according to the previous studies [23], self-help strategy (M = 3.57, SD = 0.98), accommodation strategy (M = 3.73, SD = 0.88), and approach strategy (M = 3.53, SD = 0.90) were characterized into adaptive coping. The total scores of avoidance strategy (M = 2.45, SD = 1.04) and self-punishment (M = 2.48, SD = 1.10) were categorized into maladaptive or passive coping strategy. In our study, greater scores indicated a higher level of the coping strategy type.

#### 2.2.3. Aggressive Behavior

Aggressive behavior was measured by adapting from Buss and Perry [65], comprising five items. Respondents were asked to answer on five 5-point Likert scales, “Choose the option that matches your situation from the following statements”. The items including physical aggression (“I have an uncontrollable urge to attack others”), verbal aggression (“I have an uncontrollable urge to scold others”), anger (“I will tell people frankly that I disagree with them”), and hostility (“I will lose my temper for no reason,” and “I think people are laughing at me behind my back”), anchored by 5 = “totally fit me” and 1 = “totally doesn’t fit me”. In our study, Cronbach’s α = 0.809, and greater scores indicated a higher level of aggressive behavior (M = 2.15, SD = 0.79).

#### 2.2.4. Control Variables

Demographic variables of gender (1 = men, 2 = women), ethnicity (1 = Han ethnicity, 2 = Ethnic minorities), family socioeconomic status (1 = very low, 5 = very high), and academic performance (1 = very poor, 5 = very good) was controlled in the structural equation modeling analysis.

## 3. Results

Table 1 shows the descriptive statistics of data characteristics in the present study. The demographic variables were all controlled to avoid confounding results. As results indicated (Research Question 1), information stressors ranked the highest (M = 2.98, SD = 1.10), followed by disease stressors (M = 2.71, SD = 1.10), measures stressors (M = 2.58, SD = 1.10), and environmental stressors (M = 2.02, SD = 1.00).

Figure 2 illustrates the model with the structural equation modeling results. Structural equation modeling was conducted to test the hypothesized relationships. As suggested by the literature [66,67], the structural model of the present study was statistically significant, with chi-square value (χ^2^) = 353.947, df = 110, *p* = 0.00. The statistics also showed that the structural model of the present study was a good fit with the data, and relevant fit indices were acceptable, with CFI = 0.903 (>0.9), IFI = 0.905 (>0.9), GFI = 0.939 (>0.9), AGFI = 0.905 (>0.9), RMSEA = 0.061 (<0.8). Therefore, the criteria of overall fit with the data were evaluated and the structural model achieved an adequate model fit. Then, we estimated path coefficients of the structural model as shown in Figure 2.

We evaluated each path of the model. We also tested hypothesized relationships among the variables using an associated standardized regression coefficient and *p*-value. It could be seen that the stressors of COVID-19 had a significant direct positive impact on aggressive behavior (β = 0.24, *p* < 0.001); hence, H1 was supported. The association among perceived stressors of COVID-19 and self-help strategy (β = 0.09, *p* < 0.05), avoidance strategy (β = 0.24, *p* < 0.001), and self-punishment strategy (β = 0.13, *p* < 0.01) was also significant. However, perceived stressors of COVID-19 were not significantly related to approach strategy (β = −0.04, *p* > 0.05) and accommodation strategy (β = 0.06, *p* > 0.05), implying partial support for H2. The negative relationship between approach strategy and aggressive behavior indicated that the higher level of approach strategy led to a lower level of aggressive behavior (β = −0.10, *p* < 0.05). However, self-help strategy and accommodation strategy were not related to aggressive behavior (β = −0.01, *p* > 0.05; β = −0.06, *p* > 0.05 respectively). Additionally, avoidance strategy and self-punishment strategy were found to have a significant positive influence on aggressive behavior (β = 0.20, *p* < 0.001; β = 0.32, *p* < 0.001 respectively), which was partially in accordance with H3.

## 4. Discussion

In this study we proposed a mediation model to investigate the association between stressors of COVID-19 and college students’ aggressive behaviors. The results indicate that the college students’ perceived stressors of COVID-19 directly lead to their aggressive behaviors through a survey study. In addition, the stressors of COVID-19 would indirectly affect college students’ aggressive behaviors by the mediation of coping strategies. These different coping strategies would have disparate effects on their aggression regarding the directions. This research enriches the previous literature on the relationships among stress, coping, and aggression. We also provided relevant policy recommendations for intervening and guiding college students to adopt adaptive coping strategies.

### 4.1. Theoretical Implications

First, in this study we found that college students’ stressors of COVID-19 directly lead to their aggressive behavior. In general, this finding echoes with previous research on stress and aggression. Consistent with frustration–aggression theory, perceived stress and maladaptive coping are positively associated with adjustment problems [68]. For example, to cope with cultural tensions, Mexican-American youth would engage in violence in the forms of internalized colonialism, external oppression, and actual violent behaviors (e.g., suicides, physical or sexual abuse, and gang fights) [69]; work stress and burnout would cause the workplace violence of nurses [70]; men’s violence was mainly caused by occupational and loss stressors; women’s violence in the family resulted from a wider range of stressors [71]. In particular, Ye et al. [72] posited that the stressors of COVID-19 were directly positively associated with the engagement in online aggressive behavior of Chinese college students. To our surprise, we also found that information stressors ranked the highest, while disease stressors only ranked second. This demonstrated that media indeed plays a significant role between emergencies and adolescents’ perceived stressors. We thus extend frustration–aggression theory for aggression behavior consequences in the context of the COVID-19 pandemic as well as among college student groups. All of this raises the need to further examine the impact of stress on violence and aggression in different forms of stressors and groups [73].

Second, we found that college students will indirectly affect their aggressive behavior through coping strategies. These different coping strategies will have varying effects regarding the directions on college students’ aggression. More specifically, when college students feel stressors of COVID-19, the degrees of self-help, avoidance, and self-punishment strategies will also increase. Approach strategy, as a kind of adaptive strategy, can reduce their aggressive behavior. In contrast, maladaptive strategies of avoidance and self-punishment strategies can increase their aggressive behavior. In general, this finding is in line with the general strain theory that an individual’s factor of coping strategies would influence stress consequences and play a mediation role in the relationship between COVID-19 stressors and aggressive behavior among college students. Furthermore, this study shows more evidence that adolescents and young adults are often more prone to employ maladaptive strategies than adaptive strategies when faced with stressors. This echoes with previous studies in which adaptive coping strategies are associated with better psychological adjustment, but maladaptive coping strategies are related to worse adjustment [74,75,76]. Previous longitudinal research also demonstrated that avoidance coping emerges as “a significant predictor of adolescent symptomatology across all times” when adolescents are faced with stressful events [58]. Although some coping strategies may “serve protective functions by regulating the negative emotions associated with stress, generating alternative solutions, and thus reducing the negative consequences of the stressors, others may exacerbate the effects of stress and themselves become risk factors” [77]. We thus contribute to the general strain theory by articulating the mediating effects of the two types of coping strategies. This further calls for a concrete look at the impact of different coping strategies on aggressive behavior and other problem behaviors.

### 4.2. Practical Implications

Related stakeholders need to pay attention to college students who are experiencing multiple pressures such as study, finances, and unemployment. College students are vulnerable to the impact of global pandemics and are, therefore, prone to social problems that require timely intervention [78,79]. Based on the findings of the present study, we put forward several recommendations and policy suggestions for family members, educators, social workers, and the government.

First, parents’ positive emotional transfer and psychological counseling are irreplaceable sources of confidence for children. Parents should frequently communicate with college students to help them actively cope with the psychological pressures brought by the pandemic or other stressors.

Second, schools should make better use of psychological counseling stations to relieve and regulate the pressures experienced by college students, help students avoid negative strategies to transfer the pressure to violence to venting the emotions of college students, and reduce the occurrence of campus violence and bullying. Schools should also pay attention to students who are at a disadvantage. For example, previous research has found that young people without a fixed address face more serious stressors, and they are more likely to have negative relationships with teachers and experience greater financial pressure [51]. Universities should establish regular online job fairs to communicate the supply and demand ends of the talent market and improve the employment rate of graduates.

Finally, the government and social workers should recruit volunteers with professional psychological backgrounds to carry out online psychological services and appeal to college students to actively seek psychological assistance to improve the social support offered to college students [80,81]. Professional volunteers need to know in depth whether the stress levels of college students exceeded their ability to adapt during the pandemic, understand their common coping strategies, and pay special attention to students who utilize avoidance and self-punishment strategies.

### 4.3. Limitations and Future Directions

Although this research has contributed to the research on stress, coping strategies, and aggressive behaviors among college students in the context of COVID-19, there are still some deficiencies. Researchers need to pay attention to these points when interpreting the findings of this study.

First, as with all other cross-sectional survey methods, we cannot ensure the cause effect of COVID-19 stressors even though many previous researchers found that it is the stress that makes people adopt coping strategies and results in violence. Hauser and Bowlds [77] also suggested that “stressful events during adolescence may set in motion a self-perpetuating cycle in which stress and symptomatology contribute to one another”. We hope that future researchers can adopt longitudinal investigations or experiments to confirm this causal relationship.

Second, although the samples of this study are representative of the colleges where the survey was located (e.g., the distribution of the family economic status of the students is relatively balanced), owing to the convenient sample used in this study, we cannot confirm that a survey of college students in other universities in China would reach a consistent conclusion.

Finally, some variables in this study were measured with only a single question; thus, there may be certain measurement errors. Future researchers can use questionnaires with standardized scales to investigate this phenomenon (e.g., the Adolescent Perceived Events Scale).

## 5. Conclusions

In the present study, we verified the direct effect of stressors of the COVID-19 pandemic on college students’ aggression, as well as the indirect effect via their coping strategies. College students first endured a variety of stressors as a result of COVID-19 (e.g., information stressors and disease stressors). Specifically, college students would adopt both adaptive and maladaptive coping strategies with the stressors of COVID-19 (i.e., self-help strategy, avoidance strategy, and self-punishment strategy). Although two types of maladaptive coping strategies (i.e., avoidance strategy and self-punishment strategy) positively mediated the relationship between stressors of COVID-19 and aggressive behavior, no adaptive coping strategy types could mediate the association between stressors of COVID-19 and aggressive behavior. The present research extends general strain theory and enriches the previous literature on the relationship among stress, coping, and aggression. In terms of practical implications, many lessons should be learned from the implementation of emergency remote teaching in higher education for the approximately three years of the COVID-19 pandemic. Universities, governments, and policy makers should re-imagine how to adjust and reform higher education practices in the post-COVID-19 era. The social effects of the educational reforms of supervision and blended learning should be considered and examined.

## Figures and Tables

**Figure 1 ijerph-20-03171-f001:**
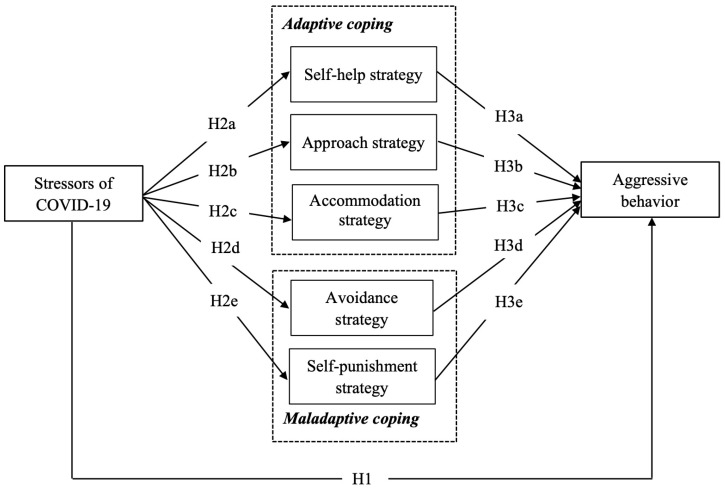
The hypothetical model of the direct and indirect effects in the present study.

**Figure 2 ijerph-20-03171-f002:**
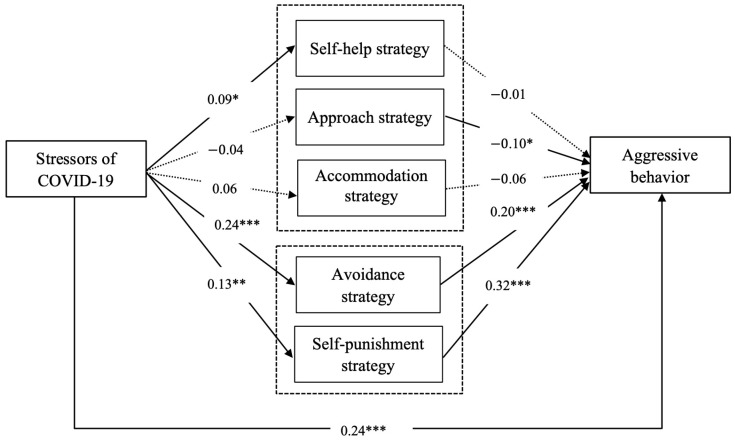
Path analyses for the direct and indirect effects in the present study. Note. Significant relationships are represented by solid lines and insignificant relationships are represented by dashed lines. * *p* < 0.05. ** *p* < 0.01. *** *p* < 0.001.

**Table 1 ijerph-20-03171-t001:** Zero-order correlations among the variables in the present study (*n* = 601).

	1	2	3	4	5	6	7	8	9	10	11
1. Gender	-										
2. Ethnicity	0.033	-									
3. Family socioeconomic status	0.048	−0.055 *	-								
4. Academic performance	0.157 ***	−0.073	0.326 ***	-							
5. Stressors of COVID-19	0.062	0.027	0.016	0.004	-						
6. Self-help strategy	0.200 ***	−0.099 *	0.199 **	0.198 ***	0.067	-					
7. Approach strategy	−0.00	−0.054	0.108 **	0.085 *	−0.070	0.344 ***	-				
8. Accommodation strategy	0.016	−0.059	0.161 ***	0.205 ***	0.009	0.351 ***	0.468 ***	-			
9. Avoidance strategy	−0.023	0.012	0.037	−0.041	0.266 ***	0.024	−0.028	0.032	-		
10. Self-punishment strategy	−0.056	0.008	−0.002	−0.019	0.165 ***	0.029	−0.085 *	0.107 **	0.345 ***	-	
11. Aggressive behavior	−0.105 *	0.013	−0.037	−0.123 ***	0.293 ***	−0.024	−0.137 **	−0.052	0.305 ***	0.383 ***	-

Note. * *p* < 0.05. ** *p* < 0.01. *** *p* < 0.001.

## Data Availability

The data that support the findings of this study are available on request from the corresponding author.

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
