# Peer review of "COVID-19 Stressors and Aggression among Chinese College Students: The Mediation Role of Coping Strategies"

_ijerph, 2023, doi:10.3390/ijerph20043171_

Round 1
Reviewer 1 Report
Thank you for inviting me to review the paper. This research study is on Chinese college students experienced excessive stressors during the COVID-19 9 pandemic. Although the introduction is particularly clear and supported, the title of the article properly represents its contents of it. However, a surfer faces the following limitation, which must be defended before publication.
There are some corrections highlighted below.
1. The abstract is particularly vague and complicated to understand. I recommend that the main concept be briefly presented, followed by the aim of the article, the method adopted, and then the main findings. presented as such.
2. The written English in this paper appears to be slightly informal; I think it is better to write in proper academic English; please revise it carefully.
3. Please elaborate in the body of the introduction because it is a brief study and the authors intend to publish this paper in a high-IF MDPI journal.
- Between 2020 and 2022, there should be a repentant literature review.
5. The conclusion is particularly poor. My recommendation is to create at least 2 subsections, one for the theoretical, managerial or other types of implications
6. I would like to suggest to authors they please cite recent research articles published on COVID-19. below link.
https://www.frontiersin.org/articles/10.3389/fpubh.2022.1035536/full?fbclid=IwAR0TpSgh4-YwQDMGuS_BE1d0MaetUTXPAUxz1tFG9kHerqzfzc_YWuueOuc
Reviewer 2 Report
Thank you very much for the opportunity to review this interesting piece of work.
The paper examined the direct effect of stressors resulting from the pandemic on college students’ aggression, as well as the indirect effect via their 12 coping strategies, using a cross-sectional survey. The topic is current and relevant for the journal.
I also agree with the methodological decisions of the authors.
At the same time there are some areas of growth I would like to highlight:
a) There is some poor paraphrasing in some paragraphs. Please check the article in Turnitin and adjust accordingly. I am not implying that there is plagiarism, but poor paraphrasing.
b) I believe the introduction needs reconsideration. Starting with violence as social issue is not a smooth introduction to the topic. I would have expected a reflection about COVID-19 and its impact on the education community. This should include a reflection about the lessons learned and how now we have a huge opportunity to redesign education and address the issues. Useful recent papers I recommend to the authors:
https://iaap-journals.onlinelibrary.wiley.com/doi/abs/10.1111/aphw.12344
https://scholarworks.waldenu.edu/hlrc/vol12/iss0/7/
c) The section about stress needs further development and detail on types of stress with a focus on college students during the COVID-19 pandemic. Otherwise I can't see strong links to the topic of the paper. There is a very recent paper (January 2023) which may be useful for this paper:
https://link.springer.com/article/10.1007/s12144-022-04214-4
What kinds of stress do we identify on college students as a result of COVID-19? For example, colleges students experience technostress due to the extensive use of technology,
https://www.mdpi.com/2078-2489/12/12/497
social isolation,
https://www.sciencedirect.com/science/article/pii/S1054139X21005036
financial stress,
https://journals.sagepub.com/doi/full/10.1177/21676968211066657
etc.
These don't need to be expanded, but the authors must name and describe them, so we understand how they lead to violence and aggressive behaviors.
d) The conclusion section needs reconsideration. It seems more like a summary, while conclusion should have a more critical and reflective approach. The conclusion should reflect on the research questions of the paper.
e) This topic is very current and useful, but I see half of the references being post 2020. I encourage the authors to enrich their reference list with more 2021 and 2022 resources.
I hope you will find my feedback useful.
Round 2
Reviewer 2 Report
Thank you so much for revising your paper and taking into consideration my comments.
The introduction provides the readers with an overview picture of the topic with a reflection on COVID-19 pandemic and its implications.
I appreciate the fact that some references have been added. I believe this is a area of growth for the paper, since there is so much research published over the last 2-3 years on this topic.
I also appreciated the restructuring of the abstract, as well as the new concluding statements.